# A Cloud Detection Neural Network Approach for the Next Generation Microwave Sounder Aboard EPS MetOp-SG A1

Salvatore Larosa [1,*], Domenico Cimini [1,2], Donatello Gallucci [1], Francesco Di Paola [1], Saverio Teodosio Nilo [1], Elisabetta Ricciardelli [1], Ermann Ripepi [1] and Filomena Romano [1]

1   Institute of Methodologies for Environmental Analysis, National Research Council (IMAA/CNR), 85100 Potenza, Italy
2   Center of Excellence Telesensing of Environment and Model Prediction of Severe Events (CETEMPS), University of L'Aquila, 67100 L'Aquila, Italy
*   Correspondence: salvatore.larosa@imaa.cnr.it; Tel.: +39-0971-427312

**Abstract:** This work presents an algorithm based on a neural network (NN) for cloud detection to detect clouds and their thermodynamic phase using spectral observations from spaceborne microwave radiometers. A standalone cloud detection algorithm over the ocean and land has been developed to distinguish clear sky versus ice and liquid clouds from microwave sounder (MWS) observations. The MWS instrument—scheduled to be onboard the first satellite of the Eumetsat Polar System Second-Generation (EPS-SG) series, MetOp-SG A1—has a direct inheritance from advanced microwave sounding unit A (AMSU-A) and the microwave humidity sounder (MHS) microwave instruments. Real observations from the MWS sensor are not currently available as its launch is foreseen in 2024. Thus, a simulated dataset of atmospheric states and associated MWS synthetic observations have been produced through radiative transfer calculations with ERA5 real atmospheric profiles and surface conditions. The developed algorithm has been validated using spectral observations from the AMSU-A and MHS sounders. While ERA5 atmospheric profiles serve as references for the model development and its validation, observations from AVHRR cloud mask products provide references for the AMSU-A/MHS model evaluation. The results clearly show the NN algorithm's high skills to detect clear, ice and liquid cloud conditions against a benchmark. In terms of overall accuracy, the NN model features 92% (88%) on the ocean and 87% (85%) on land, for the MWS (AMSU-A/MHS)-simulated dataset, respectively.

**Keywords:** neural network; microwave; cloud detection; MWS; AMSU-A; MHS

## 1. Introduction

Satellite observations are very significant information for studying clouds and their interaction with electromagnetic radiation, as well as for determining cloud properties [1,2]. Although microwaves are less affected by clouds compared to observations in the visible and infrared range, different types of clouds can affect the measured microwave radiations [3–5]. In fact, microwave observations can penetrate the whole cloud layer and are sensitive to cloud water and ice contents. Ice clouds are almost transparent below 50 GHz, except when associated with deep convective clouds, whereas liquid water clouds absorb very strongly, due to the different dielectric properties of ice hydrometeors and water drops. In case of cloud contaminated fields of view, radiative transfer simulations need more information in the input regarding cloud phase, particle size and liquid/ice cloud content. Despite this, the simulated brightness temperature often differs from the observed one, due to limitations of the cloud model and nonlinear radiation processes [5–7]. Hence, if cloud-contaminated observations were assimilated without correction, this would have a negative impact on the prediction [8–10]. It has been shown [11] that the global mean temperature in the lower and middle troposphere has a larger warming rate (about 20–30%

higher) when the cloud-influenced radiations of the advanced microwave sounding unit A (AMSU-A) are not rejected. Thus, the identification of cloud types plays a fundamental role both in the assimilation and estimation of geophysical parameters. For this reason, several methods were developed to detect cloud-affected pixels. The satellite cloud detection approaches usually rely on threshold tests or on contrast methods with multilevel decision trees [12–15].

In the last few years, machine learning algorithms based on observed spatial and spectral patterns have been employed to detect clouds [1,16–19]. In the AVHRR advanced very high-resolution radiometer pre-processing package (AAPP), a scattering index, defined by a linear regression model of the AMSU-A (channels 1-2-3-15), is used to detect cloud-affected pixels [20]. The cloud detection method in the ECMWF AMSU assimilation model [21–23] is based on a combination of a background contrast check for a window channel over land and the ocean, and a liquid water path check over the ocean. Data over land are identified as cloudy for a 50.3 GHz window channel background contrast larger than 0.7 K. Pixels over the ocean are identified as cloudy for a liquid water path (derived from window channel observations) higher than 0.2 kg/m$^2$ or for a background contrast of the 50.3 GHz window channel higher than 3 K. The difference between the observation and background (or first guess) scattering index has been used to estimate the best thresholds to filter the dataset. Qin et al. [24] proposed a new land index for cloud detection based on the overall variability of brightness temperature observations on different MHS channels in the presence of clouds. A comparison of the spatial distribution of microwave humidity sounder (MHS) cloud FOVs with clouds detected by GOES confirms a good agreement. A one-stream cloud detection method based on both the liquid water path (LWP) and ice water path (IWP) retrieved from the MHS window channels has been described by Zou et al. [9]. A cloud mask and classification based on AMSU-A/B observations have been described in [25]. In this algorithm, the visible and infrared data from the Meteosat second-generation spinning enhanced visible and infrared imager (MSG-SEVIRI) have been used to train the microwave classifier. By using this algorithm, clear, contaminated by low, medium and high clouds homogeneous pixels over the ocean and on land can be identified with a confidence level higher than 80%. Thin clouds have not been considered because, according to the authors, microwave observations are not sensitive to this type of cloud. Lindskog et al. [26] used the LWP and the scattering index in order to assimilate AMSU-A channel 6. In case of a very high LWP or enhanced scattering from large particles, AMSU-A channel 6 observations are considered contaminated by large hydrometeors. Zhu et al. [27] included optically thin clouds in the assimilation: a so-called delta-cloud liquid water ($\Delta$CLW) difference term was calculated between the cloud liquid water (CLW) using observed and simulated brightness temperatures, serving as a predictor in the radiance bias correction scheme [28]. This DCLW bias correction predictor is removed in the all-sky approach. Conversely, thick clouds are screened out using the same procedure. A new index for the detection of clouds has been developed on the basis of the differences in the response characteristics of different channels to clouds, in particular, five window and low-peaking channels (channels 1–4 and 15 AMSU-A) [29,30]. Qin et al. [31] retrieved the LWP and IWP simultaneously from collocated AMSU-A and MHS data. If either the LWP or IWP is greater than 0.02 g/kg, the corresponding AMSU-A observation is rejected as cloud-contaminated data.

Methods have been developed to detect microwave subpixel clouds using collocated moderate resolution imaging spectroradiometer (MODIS) [12] and visible infrared imager radiometer suit (VIIRS) [32] products with high spatial resolutions. Authors [33] have shown that the MODIS and VIIRS CM products can be used for subpixel cloud characterization for AMSU-A and advanced technology microwave sounder (ATMS) [34] radiation assimilation. Buehler et al. [35] presented a cloud detection algorithm that exploits the channels around the 183 GHz water vapor band. The method uses a viewing angle-dependent threshold for the brightness temperature at (183.31 ± 1.00) GHz and a threshold for the different brightness temperatures between this channel and another channel always in the

water vapor band around 183 GHz. Both the (183.31 ± 3.00) GHz and (183.31 ± 7.00) GHz channel as the other channel is studied. The robustness of this method for cloud detection is evaluated for a mid-latitude winter in a case study. Methods for detecting deep convective clouds based on optical and microwave data have been investigated. Window microwave channels can discriminate between deep convective clouds accurately, by means of differences between three water vapor channels [36]. The three water vapor channels around the 183 GHz water vapor line of the AMSU-B are also used in [37] to detect tropical deep convective clouds and convective overshooting. Cold cloud effects on satellite measurements near 183 GHz have been analyzed in [38]. Werner et al. [39] described the training and validation of an improved aura microwave limb sounder (MLS) cloud detection scheme employing an artificial NN. This algorithm is derived from collocated MLS samples and MODIS cloud products and is designed to classify clear and cloudy conditions for individual MLS profiles.

Recently, deep learning (DL) techniques have been used for many cloud detection applications [40]. DL-based algorithms can quickly and smoothly learn the relationships that exist between different objects, avoiding artificially defined thresholds and constraints to match the spectral model. DL methods are mainly applied in cloud detection algorithms based on various satellite measurements, such as the Bayesian algorithm [41,42], random forests [43–45], support vector machine [46], artificial NNs [40,47] and others. Several DL-based applications were developed such as cloud detection [43], cirrus detection and optical property retrievals [48,49] and also cloud thermodynamic phase detection for different local times based on observations from VIIRS onboard Suomi NPP (SNPP) [46].

The main objective of this work is to develop a stand-alone cloud detection algorithm based on a NN model for a new microwave sounder able to distinguish the different cloud phases. The knowledge of cloud type, especially phase, is useful when estimating the microphysical properties of clouds with physical algorithms so that the initial guess can be better constrained for more accurate solutions. Cloud detection becomes a very complex problem over a global scale because of the variety of surface emissivity and atmospheric conditions. Therefore, neural networks are well suited for the solution of this problem when approaching it globally. The NN model was developed for the microwave sounder (MWS) instrument using simulated data since observations from this sensor will not be available until the launch of EPS-SG-A1 in 2024. The model is modular so that it can also be applied to AMSU-A/MHS to evaluate its performance against real observations from instruments currently in orbit. The paper is organized as follows: Section 2 first defines the criteria for collecting the dataset (Section 2.1) and then describes the cloud detection approach (Section 2.2). Section 3 discusses the results and Section 4 draws the conclusions.

## 2. Materials and Methods

This section describes the details of the procedure, criteria and data products used for the training and evaluation of the developed algorithm (Section 2.1), data preparation (Section 2.2), model performance assessment (Section 2.3), NN configuration used for cloud detection (Section 2.4) and NN training process (Section 2.5).

### 2.1. Dataset Used

#### 2.1.1. Satellite Observed Dataset

The MWS instrument is scheduled to be onboard the first satellite of the EPS-SG series—Metop-SG A1—with a tentative launch date in 2024 [50,51]. MWS has a direct inheritance from the microwave instruments AMSU-A and MHS onboard EPS and NOAA satellites. MWS is a cross-track scanning microwave radiometer with 24 channels between 23 GHz and 230 GHz, with an antenna size of ~35 cm and a variable resolution at nadir from 17 to 40 km (see Table 1). The antenna scan speed is not constant but is accelerated/decelerated in order to maximize the Earth scene viewing time. The oxygen-band channels provide a microwave temperature profile and the channels located around the 183 GHz water vapor line provide a humidity profile. The window channels give information about the total

water vapor and cloud water column. The MWS channel at 229 GHz provides improved sensitivity for the detection of cloud ice. The window channels can provide information about precipitation, sea ice and snow coverage [52].

**Table 1.** List of MWS channels with information on radiometric noise, polarization and spatial resolution.

| Channel | Frequency (GHz) | Noise Equivalent (K) | Polarization | Resolution at Nadir (km) |
|---|---|---|---|---|
| 1 | 23.8 | 0.25 | QV | 40 |
| 2 | 31.4 | 0.35 | QV | 40 |
| 3 | 50.3 | 0.50 | QV | 20 |
| 4 | 52.8 | 0.35 | QV | 20 |
| 5 | $53.246 \pm 0.008$ | 0.40 | QH | 20 |
| 6 | $53.596 \pm 0.115$ | 0.40 | QH | 20 |
| 7 | $53.948 \pm 0.081$ | 0.40 | QH | 20 |
| 8 | 54.4 | 0.35 | QH | 20 |
| 9 | 54.94 | 0.35 | QV | 20 |
| 10 | 55.5 | 0.40 | QH | 20 |
| 11 | 57.290344 | 0.40 | QH | 20 |
| 12 | $57.290344 \pm 0.217$ | 0.55 | QH | 20 |
| 13 | $57.290344 \pm 0.3222 \pm 0.048$ | 0.60 | QH | 20 |
| 14 | $57.290344 \pm 0.3222 \pm 0.022$ | 0.90 | QH | 20 |
| 15 | $57.290344 \pm 0.3222 \pm 0.010$ | 1.20 | QH | 20 |
| 16 | $57.290344 \pm 0.3222 \pm 0.0045$ | 2.00 | QH | 20 |
| 17 | 89.0 | 0.25 | QV | 17 |
| 18 | 164.0–167.0 | 0.50 | QV | 17 |
| 19 | $183.311 \pm 7.0$ | 0.40 | QV | 17 |
| 20 | $183.311 \pm 4.5$ | 0.40 | QV | 17 |
| 21 | $183.311 \pm 3.0$ | 0.60 | QV | 17 |
| 22 | $183.311 \pm 1.8$ | 0.60 | QV | 17 |
| 23 | $183.311 \pm 1.0$ | 0.75 | QV | 17 |
| 24 | 229.0 | 0.70 | QV | 17 |

AMSU-A [53] and MHS [54] are onboard the NOAA series and MetOp European meteorological satellites, and both are the cross-track microwave radiometers. AMSU-A scans the Earth scene within $\pm 48.7°$ with 30 FOVs, while MHS scans the Earth scene within $\pm 49.44°$ with 90 FOVs (see Table 2). The FOV size increases with the scan angle, at nadir the spatial resolution is 48 km for the AMSU-A and 16 km for the MHS. The AMSU-A provides a microwave radiometer measuring scene radiances in 15 frequency channels (23–90 GHz). Channels near the 50 GHz oxygen absorption bands carry information for atmospheric temperature sounding, while window channels provide information on water vapor, surface temperature, clouds and emissivity. The MHS channels around the 183 GHz band have been designed for the humidity profile retrievals. AMSU-A channels 1 and 2 are used for the retrieval of the liquid water path over the sea and the MHS channels 1 and 2 are used for the ice water path physical retrieval over both land and sea [53,55].

The advanced very high-resolution radiometer 3 (AVHRR/3) onboard MetOp-B/C is a cross-track scanner. It has six channels between the visible and infrared regions. The MetOp-B/C AVHRR cloud cover layers (CCL) data are generated by the clouds from the AVHRR extended processing system [56] and are delivered at two different resolutions: FRAC/HRPT (1 km) and GHRR (4 km). The data are distributed through the comprehensive large array data stewardship system (CLASS) in NetCDF-4 format.

**Table 2.** As in Table 1 but for AMSU-A and MHS channels.

| Channel | Frequency (GHz) | Noise Equivalent (K) | Polarization | Resolution at Nadir (km) |
|---------|-----------------|----------------------|--------------|--------------------------|
| | | AMSU-A | | |
| 1 | 23.8 | 0.30 | QV | 48 |
| 2 | 31.4 | 0.30 | QV | 48 |
| 3 | 50.3 | 0.40 | QV | 48 |
| 4 | 52.8 | 0.25 | QV | 48 |
| 5 | $53.596 \pm 0.115$ | 0.25 | QH | 48 |
| 6 | 54.4 | 0.25 | QH | 48 |
| 7 | 54.94 | 0.25 | QV | 48 |
| 8 | 55.5 | 0.25 | QH | 48 |
| 9 | 57.290 | 0.25 | QH | 48 |
| 10 | $57.290 \pm 0.217$ | 0.40 | QH | 48 |
| 11 | $57.290 \pm 0.3222 \pm 0.048$ | 0.40 | QH | 48 |
| 12 | $57.290 \pm 0.3222 \pm 0.022$ | 0.60 | QH | 48 |
| 13 | $57.290 \pm 0.3222 \pm 0.010$ | 0.80 | QH | 48 |
| 14 | $57.290 \pm 0.3222 \pm 0.0045$ | 1.20 | QH | 48 |
| 15 | 89.0 | 0.50 | QV | 48 |
| | | MHS | | |
| 1 | 89.0 | 0.22 | QV | 16 |
| 2 | 157.0 | 0.34 | QV | 16 |
| 3 | $183.311 \pm 1.00$ | 0.51 | QH | 16 |
| 4 | $183.311 \pm 3.00$ | 0.40 | QH | 16 |
| 5 | 190.311 | 0.46 | QV | 16 |

### 2.1.2. Satellite Synthetic Dataset

MWS measurements will not be available until the launch of EPS-SG-A1 in 2024. Hence, in this work, MWS synthetic observations are simulated using the RTTOV-SCATT [57,58] radiative transfer code. The atmospheric profiles and surface data required as inputs to the radiative transfer model are provided by ERA5 [59] on a regular $0.125° \times 0.125°$ spatial grid over the entire globe. Four days have been selected (1 January, 1 April, 1 July and 1 October 2019) at the four synoptic hours (00, 06, 12, 18) in order to describe the diurnal and seasonal cycle on a global scale. The atmospheric profiles here consist of temperature, water vapor, cloud liquid content, cloud ice content, cloud fraction and rain and snow water contents. The surface data consist of the skin temperature, the wind component and the 2 m temperature and humidity. Land emissivity is from the TELSEM2 [60] atlas while ocean emissivity is from the FASTEM model [61] integrated in the RTTOV. The scattering calculation in the clouds is based on the delta-Eddington approximation and tables of hydrometeor optical properties (rain, snow, cloud liquid water and cloud ice) are pre-calculated for the required frequencies and temperatures. The optical properties are archived in sensor-specific coefficient files, including the MWS, AMSU and MHS and the observation data used in this work. Cloud water and rain hydrometeors are represented by Mie spheres and ice clouds are represented by Mie spheres and ARTS "ice habits" [62] in our simulation. The all-sky brightness temperature is calculated as the combination of independent clear and cloudy columns weighted by an effective cloud fraction. The effective cloud fraction is calculated through a hydrometeor-weighted average across the vertical profile of the input cloud fraction [63]. Although microwave scattering calculations are simplified, RTTOV-SCATT gives much more negligible errors than all other uncertainties involved and is therefore suitable for operational use [64]. The radiometric noise requirements available through the WMO observing systems capability analysis and review (OSCAR) tool (https://space.oscar.wmo.int/, accessed on 18 February 2023) has been used for all the channels. Simulated observations for the channel suites of the MWS and AMSU-A/MHS have been produced. Bias correction was applied to match the synthetic simulated values with the observations. Simulated and measured data in clear sky conditions were compared and resulting biases, (attributed to uncertainty in the

radiative transfer model, the surface emissivities, etc.), were removed from the synthetic datasets following the approaches of Harris and Kelly [65] for the AMSU-A and MHS [5]. The same correction will undergo the MWS when the measured data will be available. For example, Figures 1 and 2 show the spectral and spatial variability of simulated observations for 1 October 2019.

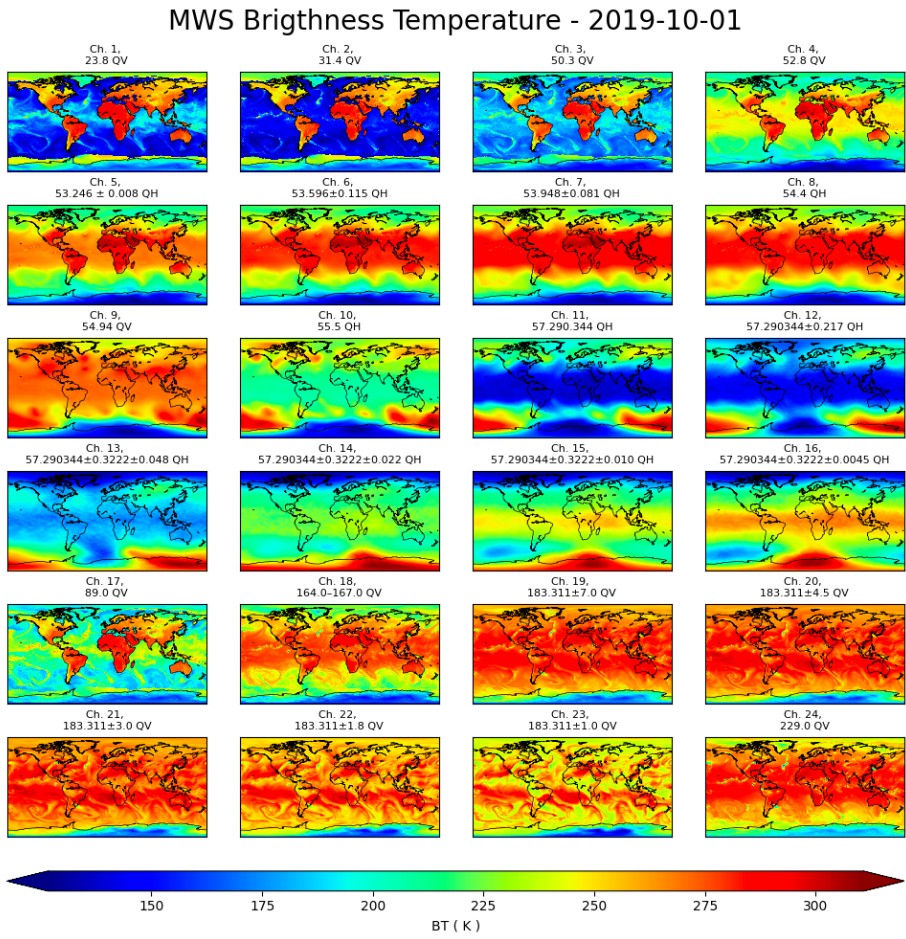

**Figure 1.** Spatial and spectral distribution of MWS synthetic dataset (1 October 2019).

## 2.2. Data Preparation

The whole dataset has been partitioned into three different subsets: (1) training, (2) test and (3) validation datasets. Training and test datasets are used to learn and assess the performance of the NN model. The validation dataset, consisting of unseen data, is used for a final evaluation of the NN performances with respect to new input data. Precisely, we divided the entire data set as follows: 70% for training, 20% for testing and 10% for validation. The dataset was preliminarily filtered to consider ocean and land surface backgrounds separately. The separation shall facilitate the classification as the surface contribution is large at some microwave channels. The ocean/land classification of the pixel was made based on the percentage of ocean or land contained. The ERA5 data have been used as reference truth for both training and validation of the NN. As mentioned, four representative days (1 January, 1 April, 1 July and 1 October 2019), each at four synoptic hours (00, 06, 12, 18), have been used.

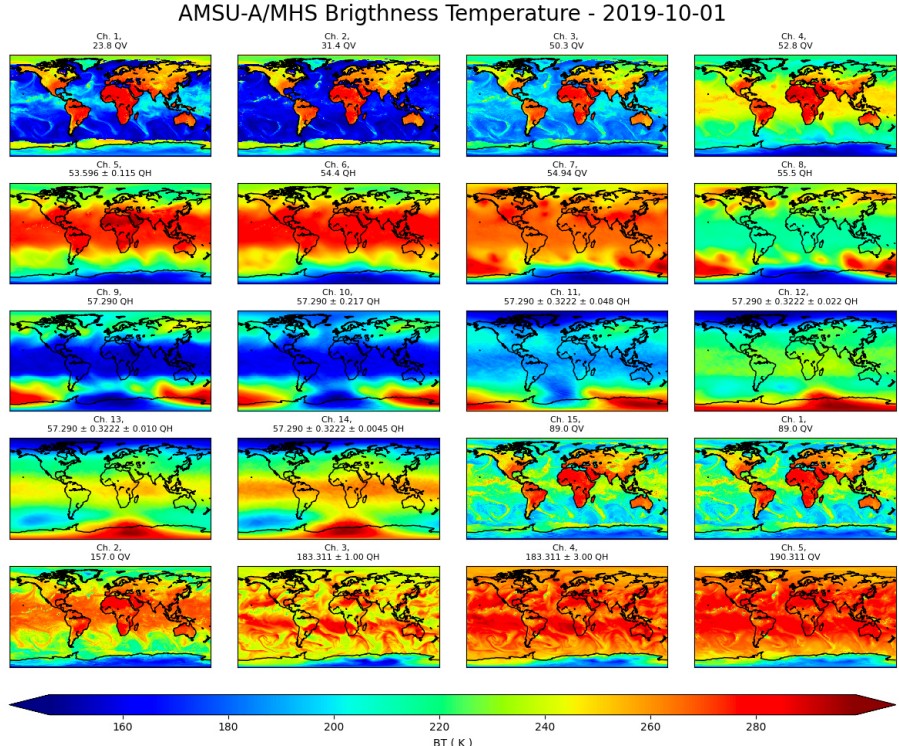

**Figure 2.** As in Figure 1, but for AMSU-A and MHS (1 October 2019).

Successively, the reference truth datasets have been processed to create a categorical labels vector that contains only the three classes to predict, e.g., clear, ice and liquid. An undetermined class has also been created for pixels that do not fall into any of the three classes defined above. Furthermore, a balanced dataset has been extracted and used as an input layer for the NN to obtain approximately the same sample number for each class and avoid potential issues with an unbalanced dataset. This pre-processing screening selects nearly 2 million (1,846,569) samples of both the MWS and AMSU-A/MHS data, which are then divided into train, test and validation subsets.

*2.3. Evaluation Metrics*

The following metrics are used in the performance evaluation of the NN based on a review of the scientific literature on cloud detection and machine learning [66]. In the following equations below, *TP*, *TN*, *FP* and *FN* denote true positive, true negative, false positive and false negative, respectively.

- The overall accuracy is calculated as the sum of the hits (correct classification) from each of the classes divided by the sum of the total points for the classification.

$$overall\ accuracy = \frac{TP + TN}{TP + TN + FP + FN} \tag{1}$$

- The precision is the ratio of true positives to all positives predicted by the model. The more false positives the model predicts, the lower the precision.

$$precision = \frac{TP}{TP + FP} \tag{2}$$

- The recall is the ratio of true positives to all positives in the data set. It measures the ability of the model to detect positive samples.

$$recall = \frac{TP}{TP + FN} \tag{3}$$

- The *F1* score is a single metric that combines precision and recall. The higher the *F1* score, the better the performance of the model.

$$F1 \text{ score} = \frac{2}{\frac{1}{recall} + \frac{1}{precision}} \tag{4}$$

- The $F_\beta$ score is the weighted harmonic mean of precision and recall and reaches its optimal (worst) value at 1 (0).

$$F_\beta \text{ score} = \left(1 + \beta^2\right) \frac{precision \times recall}{(\beta^2 \times precision) + recall} \tag{5}$$

- The area under the receiver operating characteristic curve (AUC-ROC) is a performance measurement for the classification problems. ROC is a probability curve and AUC represents the degree or measure of separability [67]. It provides a measure of the model skill to distinguish different classes. The higher the AUC, the better the model predicts 0 and 1 classes correctly. It is defined as the ratio of *TPR* against *FPR*

$$TPR = \frac{TP}{TP + FN}, \ FPR = 1 - \frac{TN}{TN + FP} \tag{6}$$

where *TPR* (*FPR*) is the true (false) positive rate.

- The Jaccard index is a measure of similarity between two sets and is related to recall and precision.

$$Jaccard \ index = \frac{TP}{TP + FN + FP} \tag{7}$$

- The Matthews correlation coefficient [68] is regarded as a balanced correlation coefficient that returns a value between $-1$ and $+1$, where $+1$ represents a perfect prediction, 0 an average random prediction and $-1$ an inverse prediction

$$MCC = \frac{TP \times TN - FP \times FN}{\sqrt{(TP + FP) \times (TP + FN) \times (TN + FP) \times (TN + FN)}} \tag{8}$$

### 2.4. Neural Network Configuration

The main framework used in this work for designing and testing the NN models is Keras 2.9.0 (https://keras.io/), a powerful Python-based library that contains numerous implementations commonly used in NN building blocks, such as layers, targets, activation functions and optimizers.

The NN model is based on a multiclass classification problem with $k$ classes, where $k = 3$, i.e., clear-sky, ice and liquid phases. For the NN, we assume that the training data are represented by $n$ features and label couples $\{(x_i, Y_i)\}_{i=1}^n$ where $x_i \in \mathbb{R}^d$ are the features (i.e., satellite observations) and $Y_i \in \{1, 2, \dots, k\}$ are the labels which represent one of the $k$ classes (i.e., clear, liquid cloud, ice cloud). In addition, it will be convenient to represent the labels as one-hot encoded vectors $y_i \in \mathbb{R}^k$ characterizing one of the $k$ classes with one-hot encoding, e.g., $y_i = e_{Y_i}$.

The $X = [x_1 x_2 \dots x_n] \in \mathbb{R}^{d \times n}$ and $Y = [y_1 y_2 \dots y_n] \in \mathbb{R}^{k \times n}$ matrices denote the features and their labels aggregated into a matrix. In our analysis, we focus on training a non-linear classifier. The bias vector that we use $b_i \in \mathbb{R}^{k \times d}$ and weight $W = [\omega_1 \omega_2 \dots \omega_m]^T \in \mathbb{R}^{k \times d}$ represent the biases and weights of this non-linear model. The correlation between the overall input/output of the classifier is a function that maps an input vector $x \in \mathbb{R}^d$ into an output of size $k$ via $x \mapsto W \times x + b \in \mathbb{R}^k$, where a training procedure is used to train, respectively, weights $W \in \mathbb{R}^{k \times d}$ and biases $b \in \mathbb{R}^{k \times d}$ [69].

A schematic drawing for a typical three-layer fully connected network is given in Figure 3. Figure 4 shows a generic *i*-th artificial neuron, showing the processing at a single given node of the NN.

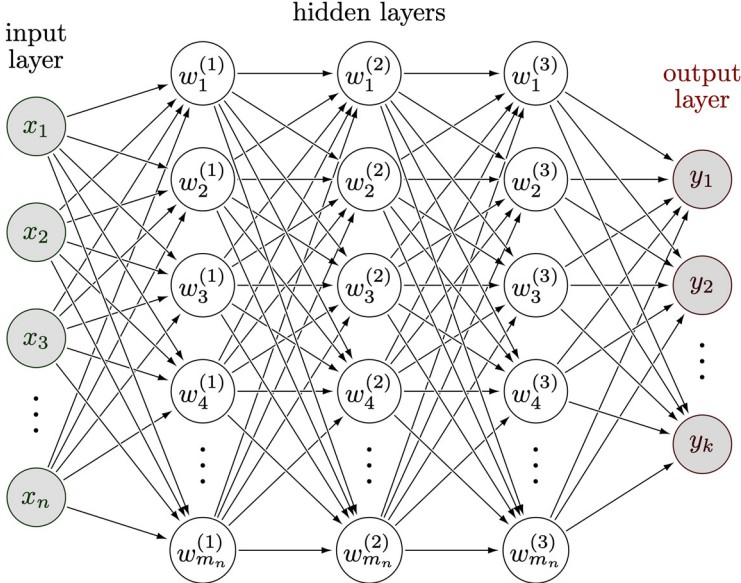

**Figure 3.** Schematic illustration of a neural network. $x_1 \ldots x_n$ represent the network inputs, $\omega_1 \ldots \omega_m$ represent the network weights and finally, $y_1 \ldots y_k$ represent the network outputs used in the NN model.

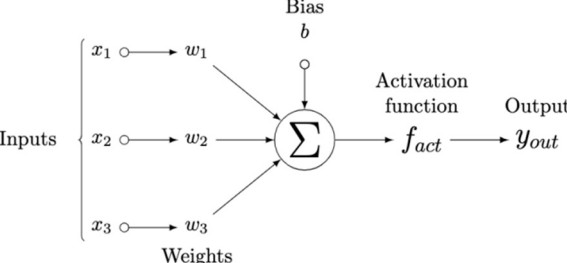

**Figure 4.** Schematic illustration of an artificial neuron.

Each single neuron model consists of a processing element with synaptic input connections and three distinct outputs. The function *f* is usually referred to as the activation function, which usually performs a nonlinear operation. Depending on the layer, different activation functions can be used in the same NN.

The function ReLU (2.9.0) [70] is used here to activate the NN units for input and hidden layers: the function returns 0 if it receives a negative input, but for any positive value it returns that value. It can thus be written as follows:

$$ReLU(x) = max(0, x) \tag{9}$$

The success of ReLU (2.9.0) can be attributed to its simple implementation, which in turn reduces the computation time of the NN model [71]. While the SoftMax (2.9.0) activation function was applied to the output layer as follows:

$$softmax(z_i) = \frac{e^{z_i}}{\sum_j^n z_j} \tag{10}$$

where all the $z_i$ values are the elements of the input vector and can take any real value, whereas *n* is the number of classes in the multi-class classifier. The denominator is the

normalization term that ensures that the sum of all output values of the function is 1, thus ensuring a valid probability distribution. The input vector is converted to a probability vector by the softmax function, where the probabilities of the individual output values are proportional to the relative size of the individual values in the same vector.

Cross entropy was chosen as the loss function of the model, whereas the adaptive moment estimation (Adam) was selected as the optimizer of the model in training [71]:

$$Loss = -\sum_{i=1}^{n} y_i \times \log \hat{y}_i \tag{11}$$

where $\hat{y}_i$ is the $i$-th scalar value in the model output, $y_i$ is the corresponding target value and $n$ is the number of scalar values in the model output.

To minimize the prediction results and label errors, the Adam optimizer and backpropagation were used to dynamically adjust the model parameters. Adam optimization is a stochastic gradient descent method based on adaptive optimization of first- and second-order moments. The optimization method is computationally efficient, requires little memory, is invariant to diagonal scaling of gradients and is well suited for large data/parameter problems [72]. For each parameter $\omega^j$

$$v_t^j = \beta_1 \times v_{t-1} - (1 - \beta_1) \times \mathfrak{g}_t \tag{12}$$

$$s_t^j = \beta_2 \times s_{t-1} - (1 - \beta_2) \times \mathfrak{g}_t^2 \tag{13}$$

$$\Delta\omega_t^j = -\eta \frac{v_t}{\sqrt{s_t + \epsilon}} \times \mathfrak{g}_t \tag{14}$$

$$\omega_{t+1}^j = \omega_t^j + \Delta\omega_t^j \tag{15}$$

where $\eta$ is the initial learning rate, $\mathfrak{g}_t$ is the gradient at time t along $\omega_t$, $v_t$ is the exponential average of gradients along $\omega_t$, $s_t$ is the exponential average of squares of gradients along $\omega_t$ and $\beta_1, \beta_2$ are the hyperparameters.

*2.5. Neural Network Model Training Process*

The NN model is defined as a multi-classification problem (clear, ice, liquid). The input layer of the NN model trained with the AMSU-A/MHS dataset is composed of 20 input units over an ocean test (15 AMSU-A and 5 MHS channels sequence) and 40 input units over a land test (15 AMSU-A and 5 MHS channels sequence and 20 emissivity values for each channel), while the NN model trained with the MWS dataset is composed of 24 input units over an ocean test (24 MWS channels sequence) and 48 input units over a land test (24 MWS channels sequence and 24 emissivity values for each channel).

In addition to the input layer, the NN has three hidden layers consisting of 100, 50 and 50 neurons, respectively, selected for the NN. The number of hidden layers and of associated neurons was set empirically initially. Then, the Keras Tuner framework was used for fine-tuning: after setting the problem to be solved, the Keras Tuner returns the ideal set of hyperparameters for a NN in terms of hidden layers, hidden units, activation function, optimizer, etc.

The ReLU activation function (Equation (13)) is used to activate the network units and the binary classification cross entropy (Equation (14)) is chosen as the loss function. The corresponding reference truth labels were derived from ERA5 profiles and used for training the network. The categorical labels of the reference dataset were converted to a one-hot vector [73].

The label of a specific value $y_i$ is a vector $v$ where every component of $v$ is zero except for the $i$-th component, which has a value of 1. For example, assuming we have some random variable $y$ that takes values from the set $S = \{clear, ice, liquid\}$, let $y_1 = clear, y_2 = ice$

and $y_3 = liquid$, then a one-hot encoding for $y$ would be: (1,0,0), (0,1,0) and (0,0,1). Since the one-hot encoding of the categorical variable levels only depends on the number of levels, one-hot encoding represents one of the techniques for categorizing variables to use in neural networks.

The Adam optimizer (Equations (9)–(12)) and backpropagation were used to minimize the prediction results and label errors, dynamically adjusting the model parameters with an initial learning rate of 0.001. In order to prevent any overfitting, the dropout layer parameter was set to 0.2, which means that 20% of the model parameters were reduced during the network training phase randomly. These components are followed by a fully connected output layer with a SoftMax (Equation (15)) activation function, which represents the final classification. Figure 5 shows the stability of performance and the trends of accuracy and loss function against the epochs.

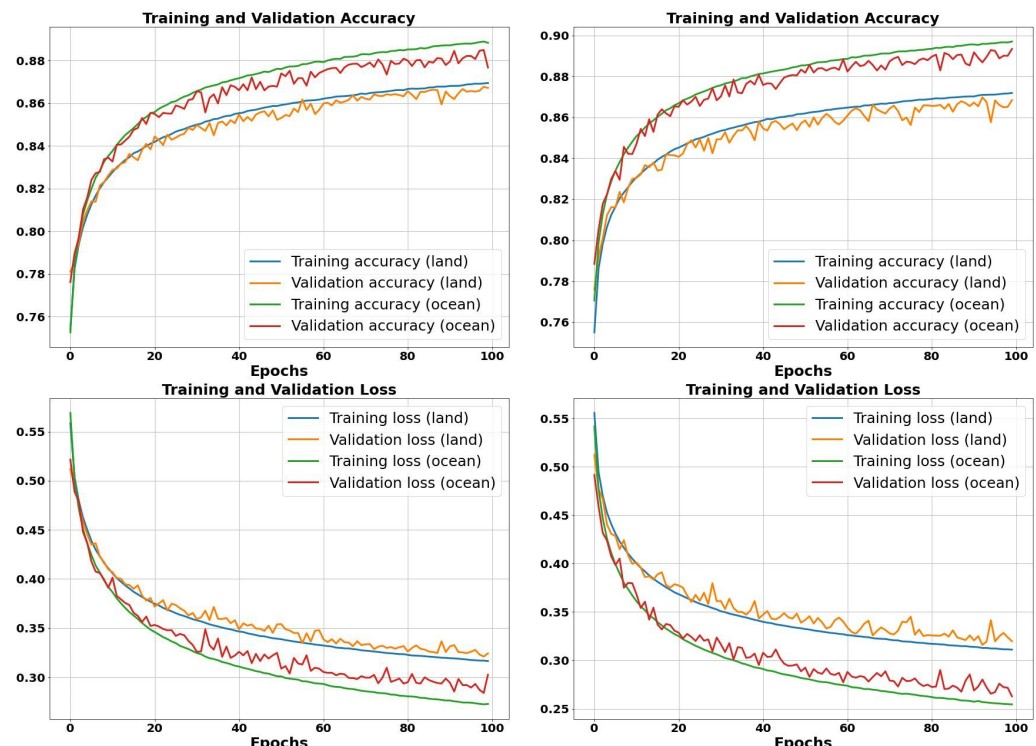

**Figure 5.** Plots of validation accuracy (**top**) and loss history (**bottom**) in NN training, for ocean and land dataset. (**left panel**): AMSU-A/MHS; (**right panel**): MWS.

Validation accuracy and loss are metrics used to evaluate the performance of a deep learning model on the validation set. The validation set is a subset of data used to evaluate the goodness of the model. For training accuracy, identical pixels are used both for training and testing, whereas for the validation accuracy the trained model identifies independent pixels that were not used in training. The model accuracy and loss along the training epochs are shown in Figure 5. The accuracy of the results shows that the model ranks the data with good accuracy, but also that the model can be further improved, e.g., by transfer learning from pre-trained models or by increasing the number of pixels within each of the classes. The peak training accuracy was 89.7% (89.2%) for training and 87.7% (86.1%) for validation, in the ocean (land) case, respectively. These models were saved and utilized for testing. Since the model was built from scratch, accuracy can be improved by using transfer learning from pre-trained models or by expanding the number of pixels in each of the classes. Independent NNs were trained and validated for different satellite scan angles from the first to the last (5° step).

## 3. Results and Discussion

In this section, we describe the results of the NN model for cloud detection from spectral microwave observations. The performance of the cloud detection algorithm was evaluated through statistical indexes as well as visual images of the pixel classifications. For the sake of clarity, this section is divided into the following: Section 3.1 illustrates the performance of the NN model from the MWS synthetic dataset and AMSU-A/MHS synthetic dataset, whereas Section 3.2 reports the validation of cloud detection retrievals from the AMSU-A/MHS measured dataset.

### 3.1. Evaluation with MWS and AMSU-A/MHS Synthetic Datasets

The performance of cloud detection using the proposed NN model has been evaluated through the metric indicators described in Section 2.3. Tables 3 and 4 report the computed statistical indicators for MWS and AMSU-A/MHS, respectively, divided into ocean and land backgrounds. The precision index, i.e., the number of pixels that are relevant out of the total pixels the NN model retrieved (perfect value is 1.00), lies in the range of 0.83–0.95 and 0.85–0.91 over the ocean, for MWS and AMSU-A/MHS, respectively, whereas over land within 0.84–0.89 and 0.81–0.91. The recall index, the number of pixels that the NN model correctly identified as relevant out of the total relevant pixels (perfect value is 1.00), is between 0.79 and 0.96 for MWS, between 0.82 and 0.93 for AMSU-A/MHS over the ocean, while between 0.78 and 0.94 and between 0.82 and 0.87 over land for the MWS and AMSU-A/MHS, respectively. The F1 score, which summarizes the predictive performance of a model by combining two otherwise competing measures, precision and recall, ranges from 0.81 to 0.96 for MWS and from 0.83 to 0.90 for AMSU-A/MHS over the ocean, while between 0.81 and 0.91 and between 0.82 and 0.89 over land for the MWS and AMSU-A/MHS, respectively. Both the recall and F1 scores can range from 0 to 1, with 1 representing perfect classification and 0 representing a model that cannot classify any observations into the correct class. The values obtained for the F1 score indicate that our model is well-suited for classification.

**Table 3.** Evaluation metrics of the cloud detection neural network model when using a synthetic dataset for MWS sensors trained and validated with ERA5 profiles. The total count of pixels N is in parentheses.

|  | Ocean (N: 184,657) | | | Land (N: 134,382) | | |
|---|---|---|---|---|---|---|
| Jaccard index |  | 77.84% |  |  | 76.96% |  |
| MCC |  | 81.39% |  |  | 80.56% |  |
| F-beta score |  | 87.18% |  |  | 86.82% |  |
| Accuracy |  | 92% |  |  | 87% |  |
| Classes | **Clear (35,549)** | **Ice (15,466)** | **Liquid (133,642)** | **Clear (49,250)** | **Ice (34,438)** | **Liquid (50,694)** |
| Precision | 0.83 | 0.86 | 0.95 | 0.84 | 0.89 | 0.88 |
| Recall | 0.79 | 0.84 | 0.96 | 0.78 | 0.88 | 0.94 |
| F1 score | 0.81 | 0.85 | 0.96 | 0.81 | 0.89 | 0.91 |
| ROC (AUC) | 0.88 | 0.91 | 0.92 | 0.85 | 0.91 | 0.94 |

Figures 6 and 7 show the predictive performance of both NN models for the MWS and AMSU-A/MHS, respectively, on the three classes (values are normalized over each column). A confusion matrix is a very popular measure used when solving classification problems [74]. The diagonal elements represent pixels that have been correctly classified. Thus, a perfect classifier has a confusion matrix with high values for the diagonal elements and zero values for the rest of the elements. The number of cases correctly predicted by our NN model is never below 80% and it is often higher than 90%. This may also be seen in Figures 8 and 9, which show the probability distribution functions of the NN model for three scene types as determined by combinations of two-class clouds with different thermodynamic phases (e.g., ice over liquid). The probability distributions show strong

peaks near 0 or 1, indicating the good performance of the NN model in distinguishing the three different thermodynamic phases included in the training and validation process.

**Table 4.** As in Table 3 but using synthetic dataset for AMSU-A and MHS sensors.

| | Ocean (N: 184,657) | | | Land (N: 134,382) | | |
|---|---|---|---|---|---|---|
| Jaccard index | | 78.24% | | | 74.19% | |
| MCC | | 81.42% | | | 77.45% | |
| F-beta score | | 87.71% | | | 85.11% | |
| Accuracy | | 88% | | | 85% | |
| Classes | **Clear (35,549)** | **Ice (15,466)** | **Liquid (133,642)** | **Clear (49,250)** | **Ice (34,438)** | **Liquid (50,694)** |
| Precision | 0.85 | 0.91 | 0.87 | 0.81 | 0.84 | 0.91 |
| Recall | 0.82 | 0.88 | 0.93 | 0.82 | 0.86 | 0.87 |
| F1 score | 0.83 | 0.89 | 0.90 | 0.82 | 0.85 | 0.89 |
| ROC (AUC) | 0.87 | 0.92 | 0.93 | 0.86 | 0.90 | 0.91 |

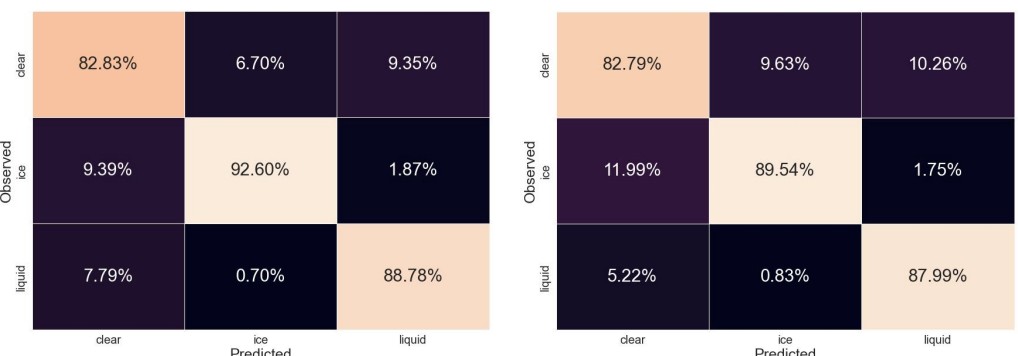

**Figure 6.** Confusion matrices for clear, ice and liquid predictions by NN models over the ocean (**left panel**) and over land (**right panel**) using the synthetic MWS dataset. The lighter boxes represent the number of pixels correctly classified for each of the classes, while the darker boxes represent misclassifications.

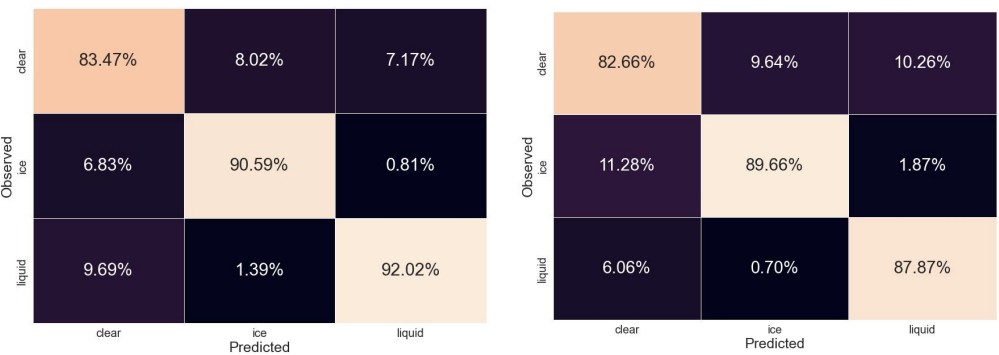

**Figure 7.** As in Figure 6 but using AMSU-A/MHS synthetic dataset.

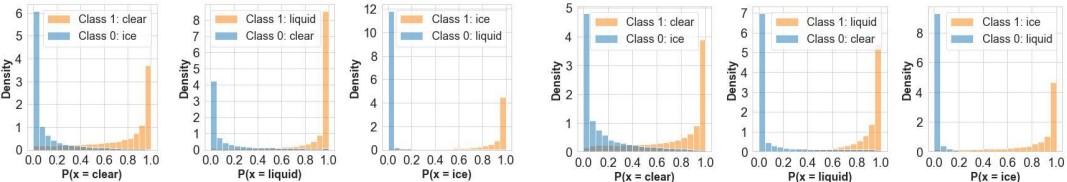

**Figure 8.** Normalized density functions of the clear, liquid and ice probabilities from the NN model using OvO (one vs. one) strategy comparing all possible two-class combinations of the synthetic MWS validation dataset. Top panels for ocean and bottom panel for land.

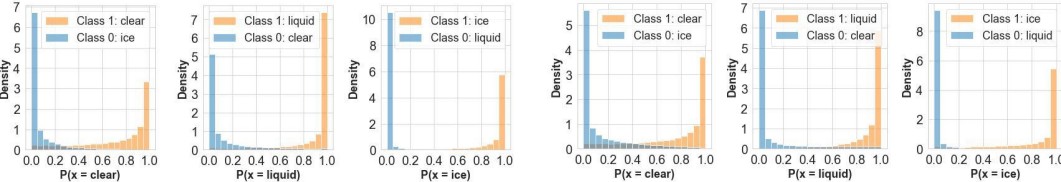

**Figure 9.** As in Figure 8 but for the synthetic AMSU-A/MHS validation dataset.

### 3.2. Detection Performance Using Observed Dataset

To evaluate the NN model performances in a real context, we used a combined observation dataset from the AMSU-A and MHS onboard MetOP-C. The considered datasets were downloaded from the EUMETSAT Earth observation portal in NetCDF format (Table 5). It consists of observations from the 15 AMSU-A channels and the 5 MHS channels during five full orbits.

**Table 5.** AMSU-A/MHS orbit and start datetime used for validation.

| Orbit Number | Start Datetime |
|:---:|:---:|
| 19375 | 1 August 2022 18:58:23 |
| 18066 | 1 May 2022 15:40:19 |
| 18935 | 1 July 2022 19:40:19 |
| 17634 | 1 April 2022 05:58:23 |
| 19370 | 1 August 2022 10:37:19 |
| 18926 | 1 July 2022 04:31:24 |
| 19369 | 1 August 2022 08:55:19 |
| 17638 | 1 April 2022 12:43:19 |

The distance between the AMSU-A and MHS footprints is negligible, being two sensors on the same platform. Each AMSU-A FOV is associated with nine MHS FOVs; the central one is concentric to the co-located AMSU-A FOV.

As a reference for the NN cloud classification, the AVHRR cloud mask has been considered. The spatial and temporal distances between AMSU-A/MHS and AVHRR/3 are negligible as the AVHRR/3 is placed onto the same spacecraft onboard MetOp-C. The cloud mask AVHRR is derived from clouds from the AVHRR extended (CLAVR-x). CLAVR-x developed at NOAA/NESDIS and UW/CIMSS generates cloud products in real time by using AVHRR data. Each AVHRR pixel has been classified into one of the following three classes: clear, liquid and ice. Collocation between AMSU-A and AVHRR uses an algorithm developed for atmospheric infrared sounder (AIRS) and moderate resolution imaging spectroradiometer (MODIS) data on NASA's Aqua satellite [75] and is an extension of the algorithms described in Li et al. [76]. This algorithm finds the closest AVHRR observation to the center of the AMSU-A footprint and performs an outward search to find all the AVHRR pixels falling within the AMSU-A footprint. Subsequently, for each AMSU-A pixel, the percentage of AVHRR pixels relative to the four classes contained within the AMSU-A pixel is calculated. According to the percentage of clear, liquid or ice, the AMSU-A FOVS is considered clear or affected by liquid or ice clouds. An AMSU-A pixel is labeled clear only if more than 85% of the related AVHRR classes are clear (different thresholds were tested and this number was found to be a good compromise). Similarly, an AMSU pixel is labeled liquid (ice) cloud only if more than 85% of the related AVHRR classes are classified as affected by liquid (ice) clouds. For land pixels, the channel emissivity values have been estimated using the TELSEM2 interpolator. The interpolator allows obtaining a reasonable emissivity for each location over the globe and for every month of the year. It should be noted that these emissivity values differ from the ones used in the simulations; while extracted from the same atlas, they refer to different data, zenith angle and geographical coordinates. Moreover, as already mentioned above (Section 2.1.2), bias correction between simulated and observed brightness radiances has been removed.

Figure 10 reports the confusion matrix, and Table 6 shows all the calculated indices. The diagonal confusion matrix values over the ocean are between 64.30% and 88.10%, and over land between 57.39% and 71.87%. The recall indices are between 0.65 and 0.94 for the ocean, and between 0.49 and 0.74 for land. The F1 score is between 0.64 and 0.76 over the ocean, and between 0.59 and 0.70 for land. Overall, Figure 10 and Table 6 indicate good performances for the NN classification. Of course, the NN classifier cannot compete with the AVHRR CLAVR-X algorithm, for several reasons, including coarser horizontal resolution and lower sensitivity to cloud particles with respect to visible and infrared observations.

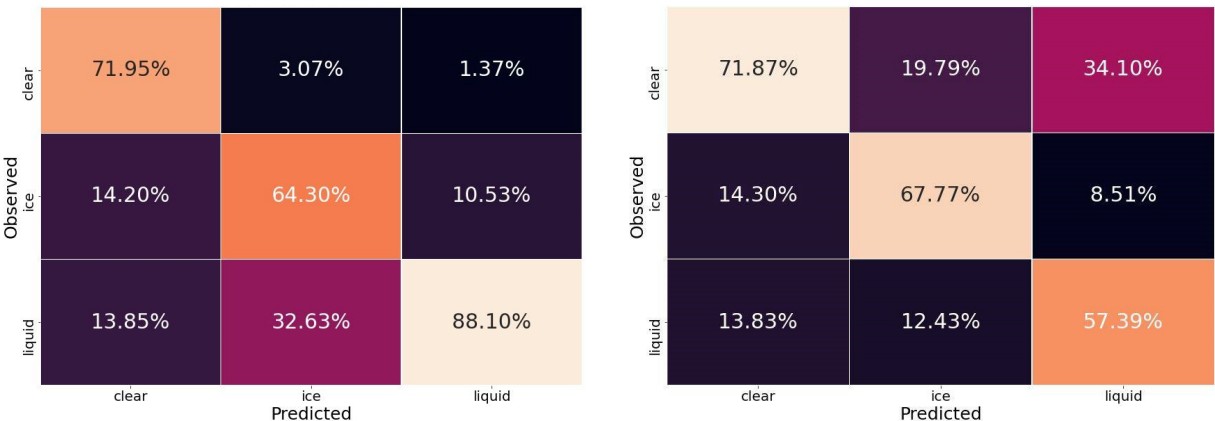

**Figure 10.** Confusion matrices for clear, ice and liquid predictions by NN models over the ocean (**left**) and over land (**right**) using the measured AMSU-A/MHS dataset. The lighter boxes represent the number of pixels correctly classified for each of the classes, while the darker boxes represent misclassifications.

**Table 6.** Evaluation metrics of the cloud detection neural network trained with a simulated dataset for AMSU-A/MHS sensors and validated with AVHRR.

|  | Ocean (N: 9333) | | | Land (N: 4135) | | |
|---|---|---|---|---|---|---|
| Jaccard index | 56.64% | | | 46.54% | | |
| MCC | 60% | | | 46.84% | | |
| F-beta score | 74.2% | | | 68% | | |
| Accuracy | 72% | | | 67% | | |
| Classes | **Clear (2729)** | **Ice (2658)** | **Liquid (3946)** | **Clear (1267)** | **Ice (1710)** | **Liquid (1138)** |
| Precision | 0.64 | 0.77 | 0.84 | 0.75 | 0.71 | 0.52 |
| Recall | 0.94 | 0.55 | 0.81 | 0.49 | 0.69 | 0.74 |
| F1 score | 0.76 | 0.64 | 0.73 | 0.59 | 0.70 | 0.61 |
| ROC (AUC) | 0.83 | 0.74 | 0.79 | 0.71 | 0.75 | 0.74 |

The trained NN model outputs were compared against the cloud phase product from CLAVR-x system, which was used as a reference image. Among all the products generated by CLAVR-x, there are also the cloud mask and phase classification for NOAA and EUMETSAT AVHRR sensors. Figures 11 and 12 show some examples of the predicted pixels by the NN model. As shown, clear-sky and cloud phases are predicted accurately over both ocean and land surfaces.

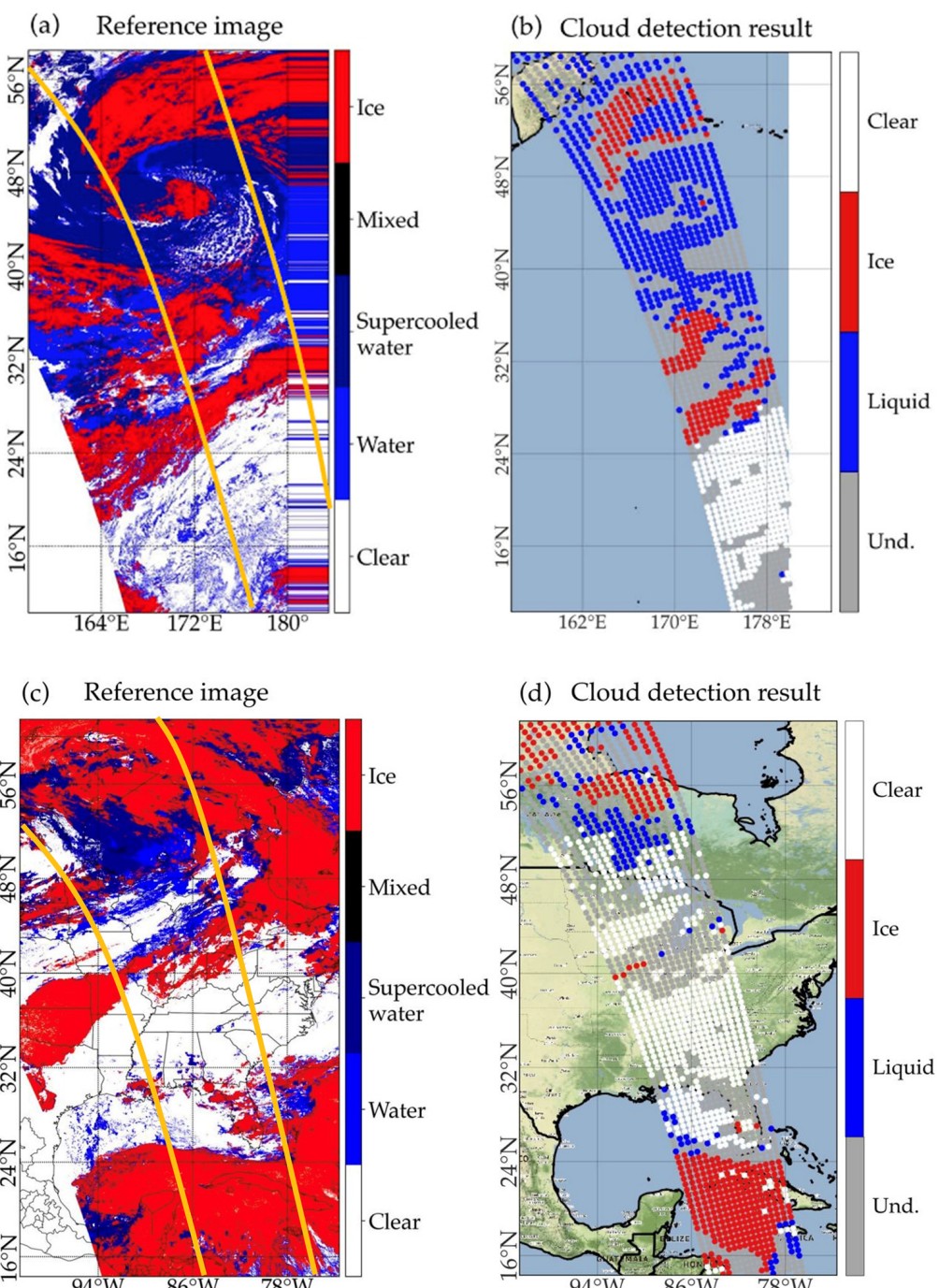

**Figure 11.** (**Top**): example for ocean background. (**a**) Pseudo-color image of the reference scene (date: 1 August 2022 18:58:23 orbit n. 19375. (**b**) Cloud detection result of this scene. (**Bottom**): example for land background. (**c**) Pseudo-color image of the reference scene (date: 1 August 2022 08:55:19 orbit n. 19369). (**d**) Cloud detection result of this scene. (Und. stands for undetermined).

Although the proposed NN model can achieve high accuracy in detecting clouds under different surface conditions (ocean and land), there are still some cases where it cannot correctly detect clouds, e.g., in regions with bright surfaces or when there is snow at high altitude/ice cover. The performances of the NN model are also closely related to the training samples. However, the model could lead to pixel misclassification if there are not enough training samples with a balanced number of pixels from the three classes (clear-sky, ice and liquid).

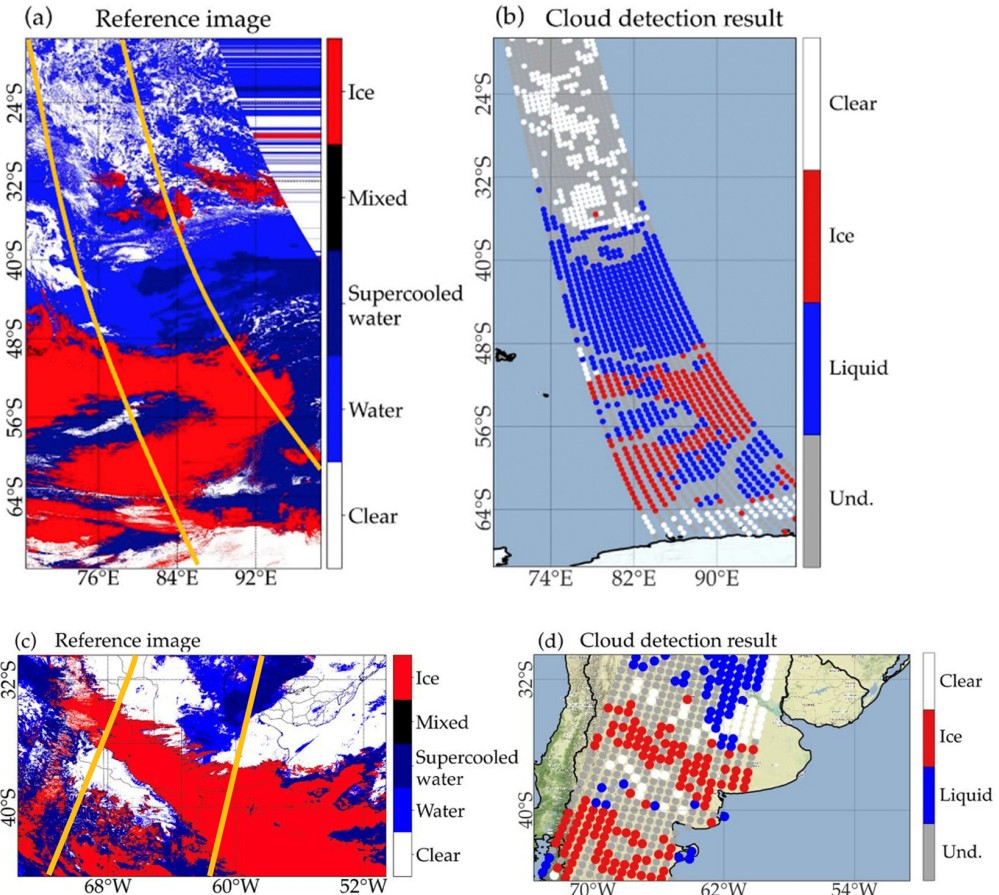

**Figure 12.** (**Top**): example for ocean background. (**a**) Pseudo-color image of the reference scene (date: 1 July 2022 19:40:19 orbit n. 18935.) (**b**) Cloud detection result of this scene. (**Bottom**): example for land background. (**c**) Pseudo-color image of the reference scene (date: 1 July 2022 04:31:24 orbit n. 18926). (**d**) Cloud detection result of this scene. (Und. stands for undetermined).

## 4. Conclusions

This study presents a NN model for cloud detection from satellite passive microwave observations. The proposed NN model detects clear, ice and liquid thermodynamic phases in a multiclass classification problem using microwave brightness temperature spectral observations over both ocean and land surface backgrounds. Over land background, emissivity values from the TELSEM2 atlas interpolator have been added as an input to the NN. The proposed NN model is designed for the MWS instrument, scheduled for launch in 2024 onboard MetOp-SG A1 as the direct successor of AMSU-A and MHS instruments, currently flying onboard EPS and NOAA satellites. A synthetic dataset of simulated MWS and AMSU-A/MHS observations has been built by processing ERA5 data with radiative transfer code RTTOV-SCAT. The synthetic dataset has been divided into three subsets for training, test and validation. Two independent NN models have been built, one exploiting MWS data and one AMSU-A/MHS data. Validation of the NN model exploiting AMSU-A/MHS has also been possible using real AMSU-A and MHS observations, against reference cloud products from AVHRR visible and infrared observations (CLAVR-x). The results clearly show that the proposed NN algorithm has good performances to detect clear, ice and liquid labels with respect to the reference. In terms of overall accuracy, the NN model has obtained 92% (87%) on the ocean (land) for the MWS dataset, while 88% (85%) on the ocean (land) for the AMSU-A/MHS simulated dataset.

In the performance evaluation, the statistical scores from Tables 3, 4 and 6 were also compared against results from the works mentioned in the introduction section of this paper, yielding similar or sometimes even better performances. For instance, Aires et al. [25]

retrieved clear sky and clouds over the ocean and land at more than an 80% confidence level, using a neural network classification method applied to homogeneous cloud pixels. Wu et al. [29] retrieved a probability of detection of the cloud fields around 84%. The number of cases correctly predicted by our NN model is never below 80% and it is often higher than 90% even under non-homogeneous pixels. Furthermore, our model, in addition to distinguishing between different cloud phases, works with all types of clouds and for non-homogeneous pixels. Therefore, the technique developed in this work can represent a valid tool for cloud detection and classification.

The encouraging results obtained with the measured dataset seem to confirm that training retrieval algorithms with simulated data are a viable option for the development of environmental products in the early deployment of future instruments, whose observations are not available at the time of development.

We plan to further improve our label dataset to make the model more accurate to address irregular and unbalanced sample distributions. At the same time, we noted that significant differences in seasonality and surface type also impact the classification ability of the NN model. For future developments, we plan to extend our NN model to other microwave sensors aboard the next-generation satellites of the EPS-SG (EUMETSAT Polar System Second-Generation) system as the Micro-Wave Imager (MWI) and the Ice Cloud Imager (ICI).

**Author Contributions:** Conceptualization, S.L. and F.R.; methodology, S.L.; software, S.L.; validation, S.L.; resources, E.R. (Ermann Ripepi); data curation, F.R.; writing—original draft, S.L. and F.R.; writing—review and editing, D.C., D.G., F.D.P., S.T.N. and E.R. (Elisabetta Ricciardelli); supervision, D.C. and F.R. All authors have read and agreed to the published version of the manuscript.

**Funding:** This research received no external funding.

**Data Availability Statement:** Not applicable.

**Conflicts of Interest:** The authors declare no conflict of interest.

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
