# Peer review of "A Cloud Detection Neural Network Approach for the Next Generation Microwave Sounder Aboard EPS MetOp-SG A1"

_remotesensing, doi:10.3390/rs15071798_

Round 1

Reviewer 1 Report

Comment to “A Cloud Detection Neural Network approach for the next generation Microwave Sounder aboard EPS MetOp-SG A1”

This study presents an algorithm based on Neural Network (NN) for Cloud Detection to detect clouds and their thermodynamic phase using spectral observations from space-borne microwave radiometers, with further evaluations against AVHRR cloud mask product. It shows that the method has great performance, suggesting the reliability of new developed algorithm. The paper is also well written. I would suggest its acceptance for publication after minor revisions.

Line 33-34, By proposing the importance of satellite cloud retrievals based on radiation, a few references might be necessary, with two suggested here, Yang et al. (2022, doi: 10.1016/j.rse.2022.112971) and Letu et al. (2020, doi: 10.1016/j.rse.2019.111583).

Line 53-54, The reference suggested above (Yang et al., 2022) also belongs to this type of method and is worthy to mention here.

Line 61-63, How is the threshold values determined or how reliable are these threshold values?

Line 63, It seems the first name of the author is mentioned here instead of last name for the reference.

Line 68, “based” should be “based on”

The first paragraph in the introduction part should be divided into 2 or even 3 paragraphs in my opinion.

Line 145, “provide” -> “providing”

Line 154-155, Personally, I would suggest changing “is” to “provides”

Line 202-203, “in order to evaluation of the …”?

Line 209-210, this is not a full sentence.

Figures 3 and 4: The meanings of the symbols should be explained in the caption.

Line 361, “divided into”

Table 3 and 4, some symbols/variables should be explained, such as N.

Reviewer 2 Report

Review.

“A Cloud Detection Neural Network approach for the next 2

generation Microwave Sounder aboard EPS MetOp-SG A1”

General remarks.

1. The introduction describes different types of clouds, different instruments and measurement methods, however, it is necessary to clearly justify why a separate cloud classification algorithm is needed. How will the solution of this problem improve the accuracy of determining other meteorological parameters?

2. The paper does not provide justification for the use of neural networks  for cloud classification. There is no comparison of the quality of the proposed method with the results of other known methods (for example, based on physical algorithms).

3. The authors do not justify why exactly 3 hidden layers were used. It is important to show that 2 layers or 4 layers will give worse results.

4. Section 2. Land emissivity from TELSEM2 [58] atlas is used to synthesize the “measured” brightness temperatures over land. As far as I understand, the same emissivity is used unchanged at the input of the neural network when solving the inverse problem. This is methodologically incorrect, since the real Land emissivity values will differ from the atlas. Therefore, in addition to the random errors associated with the noise of radiometers, it is necessary to introduce errors due to the uncertainty of Land emissivity in data synthesis.

5. Tables 1 and 2 do not indicate polarizations on different channels.

6. Lines 175-176. You should indicate which days were taken in April, July and October.

7. It is desirable to comment on whether the statistics include areas in the area of the coast, transition from land to water or vice versa. Are they included in the "land" or in the "ocean"?

8. Figures 1 and 2. A piece of the mainland of Antarctica is cut off, it must be corrected if possible. Marking "Channel #" is not convenient, you have to refer to the table on another page to understand what frequency of observation the image corresponds to. I recommend labeling images as “Chan. 1, 23.8 p”, where p is the polarization.

9. Line 351. The peak training accuracy was 89% (87%) for training and 88% (85%). Where the values of 88% (85%) are obtained from, they do not follow from the graphs in Figure 5.

10. Section 4. It is advisable to compare the accuracy of the proposed method with the accuracy of other algorithms in numerical form.

In general, the article is interesting and can be published after correcting the noted shortcomings.
